# Neutrophil Extracellular Traps (NETs) Take the Central Stage in Driving Autoimmune Responses

**DOI:** 10.3390/cells9040915

**Published:** 2020-04-08

**Authors:** Esther Fousert, René Toes, Jyaysi Desai

**Affiliations:** Department of Rheumatology, Leiden University Medical Center, 2333 ZA Leiden, The Netherlands; estherfousert@gmail.com (E.F.); R.E.M.Toes@lumc.nl (R.T.)

**Keywords:** neutrophil extracellular traps (NETs), autoimmunity, autoimmune diseases, inflammation, autoantigens

## Abstract

Following fifteen years of research, neutrophil extracellular traps (NETs) are widely reported in a large range of inflammatory infectious and non-infectious diseases. Cumulating evidences from in vitro, in vivo and clinical diagnostics suggest that NETs may play a crucial role in inflammation and autoimmunity in a variety of autoimmune diseases, such as rheumatoid arthritis (RA), systemic lupus erythematosus (SLE) and anti-neutrophil cytoplasmic antibodies (ANCA)-associated vasculitis (AAV). Most likely, NETs contribute to breaking self-tolerance in autoimmune diseases in several ways. During this review, we discuss the current knowledge on how NETs could drive autoimmune responses. NETs can break self-tolerance by being a source of autoantigens for autoantibodies found in autoimmune diseases, such as anti-citrullinated protein antibodies (ACPAs) in RA, anti-dsDNA in SLE and anti-myeloperoxidase and anti-protein 3 in AAV. Moreover, NET components could accelerate the inflammatory response by mediating complement activation, acting as danger-associated molecular patterns (DAMPs) and inflammasome activators, for example. NETs also can activate other immune cells, such as B cells, antigen-presenting cells and T cells. Additionally, impaired clearance of NETs in autoimmune diseases prolongs the presence of active NETs and their components and, in this way, accelerate immune responses. NETs have not only been implicated as drivers of inflammation, but also are linked to resolution of inflammation. Therefore, NETs may be central regulators of inflammation and autoimmunity, serve as biomarkers, as well as promising targets for future therapeutics of inflammatory autoimmune diseases.

## 1. Introduction

Known as one of the first responder cells of the innate immune system, neutrophils are described as phagocytes in textbooks that are involved in initial early host-defence responses during infection/injury. However, the discovery of neutrophil extracellular traps (NETs) has shifted the paradigm of our current understanding of neutrophil functions, and their significance during immune responses, quite drastically. Upon interaction with an invading microbe/cytokine, neutrophils release their chromatin material together with a wide range of granular enzymes to form net-like structures known as NETs [1]. NETs cannot only trap the invading pathogen but also degrade them with NET-associated proteolytic enzymes [1]. NETs are involved in numerous infectious/non-infectious diseases and are believed to be crucially involved during inflammation. While NETs are beneficial during infections, they may play a detrimental role in the case of inflammation, autoimmunity and other pathophysiological conditions. NETs accelerate the inflammatory processes by releasing a wide range of active molecules like danger associated molecular patterns (DAMPs), histones, as well as active lytic-enzymes in extracellular space, leading to further immune responses. NETs, therefore, also may serve as a potential source of auto-antigens against which the autoantibodies associated with a wide range of inflammatory autoimmune diseases are directed.

The functions and morphology of neutrophils undergo radical transformation during inflammation, injury and infection. Neutrophils migrate along vesicles by expressing a wide range of migratory protein cascades as well as start to express various pattern recognition receptors and secrete a wide range of cytokines in a process called ‘neutrophil activation’. Over the years, it has become clearer that only a fraction of neutrophils can make NETs, indicating the heterogeneity of the neutrophil population, especially during sterile inflammation [2,3] Therefore, it is important to speculate if only a specific subpopulation of neutrophils can undergo NET formation [2,4]. A distinct population of low-density neutrophils, for example, are known to be more vulnerable towards NET formation in systemic lupus erythematosus (SLE) patients [3,5], possibly explaining a link between this disease and NET formation. Interestingly, the composition of NETs may differ based on the stimuli and, therefore, the disease with which it is associated [6]. Furthermore, in certain situations, NETs also might have anti-inflammatory characteristics [7]. It is, therefore, important to characterize NETs in a disease-specific manner to understand their specific involvement during the development of autoimmunity and disease.

## 2. Composition of Neutrophil Extracellular Traps (NETs)

Neutrophil extracellular traps (NETs) formation can be triggered by a wide range of stimuli in vitro and in vivo during various pathophysiological conditions [6,8]. The protein cargo of NETs induced by different stimuli is heterogenous, making comparing research and drawing conclusions challenging. Due to this, there is an ongoing discussion about the precise mechanisms involved in NET formation, their composition and, thereby, their functional profile specifically their inflammatory/antimicrobial properties [6,9,10]. Recently, there have been new insights about how molecular mechanisms of NET formation may differ in a species specific manner [11,12] but, also based on the location of neutrophils in the blood stream or tissue, as well as local environmental alkaline or oxygen conditions [13]. Therefore, in the context of autoimmune diseases, detailed proteomic analysis of disease-specific NET protein composition (NETome) has the potential to elucidate novel mechanisms of disease onset and progression. The presence of DNase1 inhibitors in SLE)-associated NETs could potentially be demonstrated to lead to impairment of NET degradation [14]. Although disease-specific NETs may have different pathological roles, Chapman et al., recently compared the protein composition of SLE and rheumatoid arthritis (RA) NETs induced by the same stimulant, and showed that only a small number of NET proteins were significantly different between the two diseases [10]. Upon phorbol myristate acetate (PMA) stimulation, RNASE2 was higher in RA NETs, whereas myeloperoxidase (MPO), leukocyte elastase inhibitor and thymidine phosphorylase (TYMP) were higher in SLE NETs. Conversely, NETome comparison between NOX2-dependent (PMA) and NOX2-independent (A23187) showed more distinct protein profiles, irrespective of the disease background of the neutrophils [10]. PMA-induced NETs, for example, were decorated with the annexin proteins azurocidin and histone H3, whereas A23287-induced NETs contained more granule proteins, such as cathelicidin antimicrobial peptides CAMP/LL37 and matrix metalloproteinase-8 (MMP8) [10]. This finding indicates that the in vitro stimulant might be of more influence on the NETome compared to the phenotype of neutrophils. Another study, however, reported the presence of several proteins that varied between PMA–NETs of SLE, lupus nephritis (LN) and healthy neutrophils [15]. LN patient-derived PMA–NETs presented a high expression of α-enolase and annexin A1, compared to SLE PMA–NETs and healthy neutrophils. Accompanying these contradicting findings, however, it is important to take into account that most of these in vitro NETome studies are performed using non-physiological stimuli such as PMA or calcium ionophore A23187 [10,15]. Physiological stimulants might affect protein composition of NETs differently. More research and more critically designed experiments are needed in this direction to understand the role of NETs and the stimuli required to form them during autoimmune responses.

## 3. Contribution of Neutrophil Extracellular Traps (NETs) to Autoimmunity

Since neutrophils and NETs are present in abundance at the inflammatory sites of various autoimmune diseases, their active involvement in driving autoimmune responses is plausible. Recent evidences from various research groups indeed suggest that (the impaired clearance of) NETs themselves, as well as interactions of NETs with other immune cells could be very crucial for the development of ‘autoimmunity’ and breaking of self-tolerance (Figure 1). These various aspects associating NETs to autoimmunity are discussed in detail below.

### 3.1. Neutrophil Extracellular Traps (NETs) as a Source of Autoantigens

Various neutrophil granular enzymes, together with decondensed chromatin, are present on NETs besides the anti-microbial proteins captured by them. Nonetheless, many of these NET-associated proteins are of “autoantigenic nature” in various rheumatologic diseases (Table 1). Examples include NET-associated MPO and proteinase 3 (PR3) enzymes as major autoantigenic targets of anti-neutrophil cytoplasmic antibody (ANCA)-associated vasculitis (AAV); NET-derived extracellular nucleic acids and dsDNA as the targets of SLE autoantibodies; citrullinated proteins namely citrullinated histones as neoepitopes for anti-citrullinated protein antibodies (ACPA) in RA [16,17,18]. These NET-associated autoantigens and their contribution are discussed in a disease-specific manner below. It is noteworthy, despite numerous evidences, that the exact mechanisms of how exactly NET autoantigens drive autoimmunity are still in a pre-mature state. There are several missing links and a lack of knowledge so further research is required to define common mechanisms of NET-derived autoimmune responses. Evidences from different in vitro and in vivo studies currently support a theory of what we propose to be a ‘vicious loop of inflammation and autoimmunity’. It implies that inflammation-derived NET-components could be a source of autoantigens for autoantibody production. These autoantibodies and associated immune complexes in return may further induce NET formation, leading to a self-amplifying loop of autoimmune-inflammation [19].

### 3.2. NETs as Complement Activators

Complement activation and consumption is a hallmark of SLE [54]. Lower levels of complement factors C3 and C4, indicating classic complement consumption, was found in sera of a subset of SLE patients with impaired NETs clearance [26]. Seen in AAV, on the other hand, activation of the alternative complement pathway by NETs was demonstrated [25,55]. Tumour necrosis factor alpha (TNF-α)-primed neutrophils stimulated with ANCA lead to C3a, C5a and C5b-9 generation and could be due to granular protein properdin present on NETs [25]. Also, neutrophils stimulated with an anti-MPO antibody generated complement factors C5a and C3d, but not C1q [55]. This indicates that NETs are involved in different complement pathways in specific autoimmune diseases.

### 3.3. NETs as Damage-Associated Molecular Patterns (DAMPs)

NET-derived products could serve as damage-associated molecular patterns (DAMPs) to initiate further inflammatory response. Particularly, the presence of extracellular histones is perceived as a DAMP [56]. Other DAMPs secreted with NETs are the high-mobility group box 1 (HMGB1) and LL37. NET-derived HMGB1 is associated with SLE [36,37], but not with AAV [36]. Contrarily, HMGB1 contributed to ANCA-induced NET formation [57]. How NET-associated DAMPs may further contribute to tissue injury and inflammation needs to be investigated in detail for autoimmune-inflammatory diseases.

### 3.4. NETs as Inflammasome Activators

Inflammasomes are key players in the activation of inflammatory responses and NETs can mediate their activation. PMA–NETs activate caspase1 leading to nucleotide-binding oligomerization domain-like receptor protein 3 (NLRP3) inflammasome activation in macrophages of SLE patients [58]. Activation of inflammasome led to secretion of interleukin (IL)-18 and IL-1β. Besides being pro-inflammatory, these cytokines also can induce further NET formation. Adult-onset Still’s disease neutrophils, stimulated with PMA, also were found to activate NLRP3 inflammasome through caspase1 [59]. NET formation itself was found to be dependent on noncanonical inflammasome activation, as extrusion of NETs by neutrophils exposed to lipopolysaccharide (LPS) or Gram-negative bacteria relied on activated caspase11 and activated gasdermin D (GSDMD) [60,61]. Together, this implies that NET formation may be both dependent on inflammasome activation and also be able to activate inflammasome.

## 4. Impaired Clearance of Neutrophil Extracellular Traps (NETs)

NETs contain various active enzymes and DAMPs such as extracellular histones [10]. It is, therefore, important to degrade the NETs by phagocytic cells like macrophages [62]. DNase1 plays an important role to degrade NETs in physiology [14]. After NETs are pre-processed by DNase1, macrophages can ingest NETs for degradation [62]. PMA–NET degradation by macrophages takes place in lysosomes [63]. However, PMA–NET degradation in macrophages also occurs in the cytosol by three prime repair exonuclease (TREX1) enzymes (or DNaseIII), as well as extracellularly by DNase1L3 secreted by dendritic cells (DCs) [62]. PMA–NET clearance by healthy donor macrophages has been described as immunologically silent and without secretion of proinflammatory cytokines [63].

Interestingly, degradation of NETs is found to be impaired in several autoimmune diseases, such as SLE [14,26,64] and AAV [65]. Hakkim et al., reported that reduced DNase1 activity in a subset of SLE patients dysregulated clearance of NETs [14]. Furthermore, SLE-associated NETs activate complement, leading to deposition of C1q on NETs which further inhibits DNase1 activity [26]. A mutation in the DNase1L3 gene also is associated with a familiar form of SLE [66]. Additionally, PMA–NET uptake by monocyte-derived macrophages [67] and LPS–NETs cleared by SLE macrophages [68] led to a pro-inflammatory response, suggesting that clearance of activated NETs in autoimmune diseases also may amplify the inflammatory response. Compromised NET clearance generates the accumulation of active-NETs at inflammatory sites; leading to more inflammation and prolonged presence of NET autoantigens [69]. This could possibly break self-tolerance and, thereby, intensify the existing autoimmune response.

## 5. Interplay of Neutrophil Extracellular Traps (NETs) and Other Adaptive Immune Cells

### 5.1. NET and B Cells

NET formation by splenic neutrophils, but not circulating neutrophils, has been found to induce immunoglobulin class switching, hypermutation and secretion by activating c [70]. Also, type I interferons (IFN) α and B cell factors produced by bone marrow neutrophils have been shown to inhibit early B cell development and expansion in SLE, leading to possibly enhanced recruitment of autoreactive clones into the mature B cell repertoire [71]. Conversely, circulating neutrophils activated with ANCA caused the release of a B lymphocyte stimulator and promoted B cell survival [72]. Recently, it also was shown that LL37-DNA complexes in NETs can directly trigger self-reactive memory B cells to produce anti-LL37 antibodies in SLE [41]. Self-reactive B cells generally do not react to DNA, however, if the DNA originates from NETs in the form of LL37-DNA complex, it can trigger B cell activation in a toll-like receptor 9 (TLR9)-dependent manner [41]. Notably, SLE-immune complexes were found to induce NET formation recently [36]. Thus, it is conceivable that B cell-derived immune complexes may activate neutrophils by binding to the Fc gamma receptor IIIb (FcγRIIIb) and induce further NET formation [73], causing a feedback loop of autoimmune responses and disease progression.

### 5.2. NETs and Antigen Presenting Cells

IFNs play a key role in the pathogenesis of autoimmune diseases and are described as a hallmark for SLE [74]. Plasmacytoid dendritic cells (pDC), the main producers of IFN, can be activated by NETs [40]. Specifically, proteins LL37 and HMGB1 form complexes with NETs DNA, which further facilitates uptake and recognition of DNA by pDCs and increases production of IFN-α [42]. In g a RA mouse model, it was shown that T cell response in the presence of NETs is mediated through DC activation [75]. Spontaneous mouse and human NETs could directly activate DCs to induce costimulatory molecules CD80 and CD86, and proinflammatory cytokine IL-6. Furthermore, mouse NETs-treated DCs promoted CD4+ Th cell immune response to secrete INF-γ and IL-17. PMA–NETs activated DCs, on the other hand, were unable to activate CD4+ T cells [76].

NETs also can activate another type of antigen presenting cell (APC): macrophages [77]. However, prolonged exposure to NETs led to damage of the mitochondrial membrane of the macrophage, indicating macrophage death [77] Remarkably, the presence of LL37 and C1q on NETs increased activation of macrophages by RA NETs, leading to secretion of pro-inflammatory IL-8, IL-6 and TNF-α [7]. Activation of inflammatory macrophages also was seen in adult-onset Still’s disease [59]. Contrary, strongly activated macrophages exposed to NETs mediated an anti-inflammatory response with increased IL-10 secretion and inhibited IL-6 secretion [7].

### 5.3. NETs and T Cells

Recently, an indirect interaction between NETs and T cells was found in the joint of an RA patient [78]. Specifically, NETs were shown to be taken up by fibroblasts, which then presented NET-associated citrullinated peptides to T cells. Also, a direct interaction between PMA–NETs and T cells was proposed [76]. NETs directly prime CD4^+^ T cells and lower the activation threshold. Therefore, T cell response to specific antigens and suboptimal stimuli is increased, which would otherwise not lead to activation of resting T cells. Interestingly, this T cell priming by NETs seems to involve T cell receptor (TCR) mediation, but is not TLR9-dependent [76]. Direct interaction also was found between SLE low-density granulocytes (LDGs) and T cells [79]. Activation of T cells by SLE LDGs induced the release of proinflammatory cytokines INF-γ and TNF-α. This induction of proinflammatory cytokines was not detected by normal density granulocytes [79]. The study investigated in vitro activation of T cells by LDGs, not NETs. Considering that LDGs in SLE are known to form NETs [5], possibly NETs could play a role in the T cell activation.

## 6. Neutrophil Extracellular Traps (NETs) in Various Autoimmune Diseases

### 6.1. Rheumatoid Arthritis (RA)

RA is a chronic systemic disease characterized by joint inflammation and bone destruction. Being an autoimmune disease, a distinct feature of RA is the presence of specific autoantibodies against post-translationally modified proteins known as anti-modified protein antibodies (AMPA) in serum and synovial fluid samples of RA patients [80] Examples of AMPA include antibodies against post-translationally modified proteins like citrullinated proteins (ACPA), antibodies against protein carbamylation (anti-CarP), and antibodies against acetylated proteins (AAPA) [81,82,83] The presence of autoantibodies has been associated with disease progression and pathogenesis. ACPA can bind to citrullinate proteins to form immune complexes, inducing a pro-inflammatory response which further includes complement activation [84,85], for example. Interestingly, AMPA (for example ACPA) can be present in circulation for several years before RA onset [86]. These clinical observations indicate that systemic autoimmunity and development of autoimmune disease are not coupled, and factors driving the transition from pre-existing autoimmunity to RA pathogenesis need to be determined.

NET formation is strongly associated with inflammatory RA [18]. NETs extrude novel autoantigens, such as citrullinated histones, which can promote the autoimmune response in RA [18,32]. Certainly ACPAs are reported to recognize autoantigens on NETs [7,30]; in particular citrullinated histones [31]. The NET-protein MPO was found elevated in RA synovial fluid (SF), skin and rheumatoid nodules, indicating that NETs are present in these inflamed areas (joint/synovium). Neutrophils of RA patients show increased spontaneous NET propensity, compared to healthy control neutrophils in vitro [18,87]. NET propensity increases when neutrophils are stimulated with RA SF and ACPA RA serum [18,87]. Several research groups also have reported the potential of ACPA to drive NET formation in vitro, demonstrating the inflammatory potential of ACPA [18,87]. Elevated MPO–DNA complexes [88] and cell-free nucleosome levels [89] are detected in RA serum. These serum levels of cell-free nucleosome in RA patients also correlate with clinical parameters, such as C-reactive protein (CRP) and positivity for rheumatoid factor (RF) and ACPA [89]. The MPO–DNA complex level also correlated with the ACPA level in RA patient sera samples [88]. Thus, NETs and NET-derived products could serve as a biomarker for RA disease activity. Besides potential biomarkers, NET and NET-derived products also may be promising therapeutic targets for inflammatory RA (Table 2).

### 6.2. Anti-Neutrophil Cytoplasmic Antibodies (ANCA) Vasculitis (AAV)

AAV is a systemic autoimmune disorder, characterized by inflammation and destruction of small vessels in various organs. Presence of autoantibodies against MPO and PR3 are the key hallmarks that are believed to trigger the disease response.

PR3 is expressed on the membrane of resting neutrophils while MPO is stored within the granules of the neutrophils [122]. Their expression increases when neutrophils are activated by cytokines [123]. NETs also are decorated with MPO and PR3, as shown by several immunofluorescence studies on in vitro NETs and in vivo NETs in necrotizing lesions of AAV [17]. Colocalization of DNA, MPO and PR3 in kidney tissue of small-vessel vasculitis (SVV) glomerulonephritis patients, for example, indicates the presence of NETs and ANCA antigens in inflamed tissue [17]. MPO release in the presence of NETs also was found in kidney biopsies of MPO–ANCA-associated glomerulonephritis patients [49,50]. NETs not only were found in the kidneys of AAV patients, but in other types of vasculitis as well. Nerve samples of ANCA-associated vasculitic neuropathy, for instance, had citrullinated histones and MPO identified as NETs [47]. One recent study suggests that NET-protein neutrophil elastase can digest ANCA in a time-dependent manner, leading to pauci-immune lesions [124].

The presence of NETs in AAV patients is evident, however their potential as biomarkers for AAV disease activity is inconclusive. Regarding the serum of active AAV patients, higher MPO–DNA levels were detected compared to AAV patients in remission [125]. Another study, however, did not find a difference between the serum MPO–DNA levels in active and remissive AAV [126].

While it is becoming clear that NETs may be a crucial source of ANCA-autoantigens, some studies suggest that binding of ANCA to their target antigens, PR3 and MPO, further activate the neutrophils. NETs are released in response to ANCA stimulation [17]. A recent study, however, contrasts these findings. NET formation in vitro, through stimulation of healthy neutrophils with AAV serum, did not correlate with serum levels of ANCA [127]. Also, after depletion of immunoglobulin (Ig)-G and IgA in serum, NET formation remained unchanged, raising the question whether ANCA actually influences NET formation [127]. Added to being antigenic in nature, NETs also influence AAV disease progression by directly damaging vessels through cytotoxicity of NET-associated histone release [56]. NET-induced endothelial cell damage can be prevented when NETs are degraded with DNase1 [55]. Intravenous immunoglobulin (IVIG) is a potential treatment for AAV patients [121]. Intriguingly, the amount of PMA-induced NETs was significantly lower when neutrophils were pre-treated with IVIG before exposure to PMA [121]. Moreover, the inhibitory effect of IVIG on NETs, as well as decreased ANCA titers and pulmonary haemorrhage, also was seen in peritoneal tissue of MPO–AAV rats treated with IVIG [121]. To conclude, the presence of NETs and their autoantigenic properties in AAV patients is well-demonstrated. Therefore, NETs could represent informative biomarkers for disease diagnosis and targets for future therapeutics (Table 2). Conversely, the role of ANCA-mediated NET induction is not yet fully convincing. Further research is needed in understanding the interplay of ANCA and NETs in the pathophysiology of AAV.

### 6.3. Systemic Lupus Erythematosus (SLE)

Systemic lupus erythematosus (SLE) is a chronic autoimmune disease attacking healthy tissue across the body. SLE can manifest in milder forms, in skin and joints, while more serious manifestations impair functions of the kidneys and central nervous system. Regarding the kidneys, autoantigens and autoantibodies can deposit as immune complexes causing severe lupus nephritis. SLE progression is a result of autoantibodies against nucleic acids and dsDNA. These autoantigens can be sourced from apoptotic and necrotic material [128]. Clearance of this material is thought to be defective in SLE [129].

NETs are known to be a central source of SLE autoantigens [5,130,131,132,133]. The persistence of NETs also could extend the exposure time of autoantigens, due to impaired NET clearance and thereby contribute to SLE pathogenesis [14,133,134]. Removal of NETs by DNase1 was shown to be malfunctioning in in vitro NETs stimulated with SLE serum [14]. This finding indicates that DNase-1 inhibitors may either be present in NETs or (auto) antibodies bound to NETs could inhibit NET-degradation by protecting against proteases. Studies in SLE patients on the relation between NET degradation and disease manifestation support this hypothesis. SLE patients with impaired NET degradation were found more likely to develop lupus nephritis, compared to SLE patients with functioning NET degradation [14,135]. Impaired NET degradation also correlated with high levels of SLE-associated autoantibodies [135]. Other effects of nondegraded NETs are activation of the complement system, thereby driving inflammation forward [26], as well as priming other neutrophils into NET formation [130]. Notably, SLE patients possess a distinct set of neutrophils, called low-density granulocytes (LDGs) [3,136,137]. These specific neutrophils have an increased spontaneous NET propensity, compared to other neutrophils of SLE patients [5,29,134]. Endothelial damage is a common symptom of SLE and may be due to the actions of NETs [5], particularly those formed by LDGs [45]. A recent study characterized two subsets of SLE LDGs, CD10^+^ LDGs and CD10^–^ LDGs [138]. The phenotype of CD10^–^ LDGs display a more immature stage of neutrophil differentiation and is less prone to spontaneous NET formation, as compared to CD10^+^ LDGs [139]. Together, this points to the importance of NETs in SLE pathogenesis and how impaired clearance of NETs may prolong the (auto)inflammatory effects of NETs.

### 6.4. Antiphospholipid Syndrome (APS)

Antiphospholipid syndrome (APS) is an autoimmune disease that often occurs together with SLE, but also presents itself as a primary disease. Autoantibodies targeting phospholipids and phospholipid binding proteins, such as β2-glycoprotein 1 (β2-GP1) are the hallmarks of APS [140]. Particularly, APS patients showcase an increased risk of thrombosis. Thrombi consist of platelets and neutrophils, but also NETs, suggesting the possible role of NETs in APS development [141]. Furthermore, neutrophils from APS patients were shown to display enhanced spontaneous NET release [134,142]. NET response might be mediated through activation of TLR4 [100,142], together with reactive oxygen species (ROS) [100,142,143]. Remarkably, IgG purified from APS patients induced NET release from healthy neutrophils [142]. Furthermore, NET release did not correlate with APS disease activity, but rather with the presence of autoantibodies [134]. An increase in NET release in response to autoantibodies could be explained by the presence of β2-GP1 on the surface of neutrophils [142]. Several studies observed anti-β2-GP1 to induce NET formation in healthy neutrophils [100,142,143]. Evidences suggest that NETs may directly mediate APS pathology by featuring as a scaffold for platelets to aggregate [144], possibly by presenting a tissue factor [53,100]. Moreover, activated platelets can stimulate neutrophils to NET formation by releasing HMGB1 [141]. Accordingly, this highlights a central role for NETs in thrombosis and, thereby, APS development.

### 6.5. Multiple Sclerosis (MS)

Multiple sclerosis (MS) is a disease of the central nervous system with a strong autoimmune character. Few studies have been performed on the role of NETs in MS pathogenesis [145]. However, in a MS rodent model it became apparent that NETs could be of importance in MS, as depletion of MPO, attenuated rodent MS phenotypes and restored the blood–brain barrier integrity [146]. Elevated MPO–DNA complexes also were detected in the MS serum of patients, but these levels did not correlate with disease activity [147]. Within MS patients, differences in MPO–DNA complex levels correlated with the patient’s gender, suggesting that NETs may underlie gender-specific differences in MS pathogenesis. Moreover, LDGs, CD14^-^CD15^high^, were found in peripheral blood of MS patients, at levels comparable to SLE patients [148]. Whether the propensity of MS LDGs to form NETs differentiates from normal density neutrophils has not been investigated yet.

### 6.6. Anti-Inflammatory NETs

While NETs clearly enhance the inflammatory response in various ways, recent evidence suggests that they also may have anti-inflammatory properties or induce an anti-inflammatory response. Found in the presence of PMA–NETs, LPS-activated macrophages act in an anti-inflammatory manner [7]. A downregulation of IL-6 and an increase in IL-10 secretion by these macrophages shows the anti-inflammatory potential of NETs. This effect was enhanced in presence of C1q and LL37 [7]. Viable and apoptotic neutrophils also mediate an anti-inflammatory response in macrophages through NF-kB signalling suppression [149], suggesting that an anti-inflammatory response may be mediated by the type of neutrophil cell death. Certain proteases in NETs also could contribute to the anti-inflammatory potential of NETs. Regarding gout, where NETs are aggregated in the synovial joint, monosodium urate crystals (MSU)-induced aggregated NETs were found to degrade inflammatory mediators, such as IL-1β, IL-6 and TNF [150]. NET-proteases also could act on NET-proteins that are autoantigens [151]. Shown in the presence of the neutrophil protease inhibitor phenylmethylsulfonyl fluoride (PMSF), autoantigens in PMA–NETs remain present. While without PMSF, PMA–NET proteases degrade NET autoantigens. More research is needed toward understanding the anti-inflammatory role of NETs and their association with autoimmunity.

## 7. Conclusions and Future Directions

Neutrophils and NETs are present in abundance at the inflammatory sites of various autoimmune diseases and play an active role during the development and persistence of autoimmune responses. As a potential source of autoantigens and activators of immune-cells, NETs could be crucial attributors to the development of ‘autoimmunity’ and the breaking of self-tolerance. Further research is required to understand the physiology and responses of NETs in a disease-specific manner. Therefore, the use of physiological disease-specific stimulants to induce NETs in vitro, an improved understanding of the disease-specific NETome, as well as their impact on other immune cells and inflammation will be crucial for future research. These efforts would provide a substantial basis to target NETs as promising biomarkers or therapeutics to treat inflammatory autoimmune diseases.

## Figures and Tables

**Figure 1 cells-09-00915-f001:**
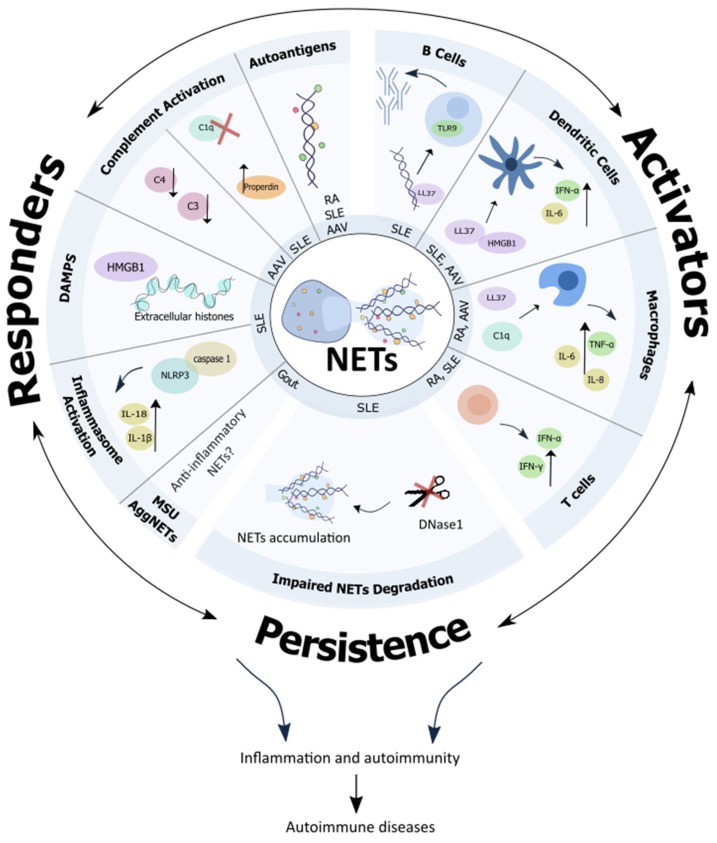
Neutrophil extracellular traps (NETs.) take the central stage in driving autoimmune responses. Abbreviations: AAV: Anti-neutrophil cytoplasmic antibodies associated vasculitis, ANCA: Anti-neutrophil cytoplasmic antibodies, C1q: complement factor 1q, C4: complement factor 3, C4: complement factor 4, DAMPs: damage associated molecular patterns, HMGB1: high mobility group box protein 1, IFN-α: interferon alpha, IFN-γ: interferon gamma, IL: interleukin, LL37: cathelicidin antimicrobial peptides, MSU: monosodium urate crystals, NETs: neutrophil extracellular traps, NLRP3: nucleotide-binding oligomerization domain-like receptor protein 3, RA: rheumatoid arthritis, SLE: systemic lupus erythematosus, TLR9: toll-like receptor 9, TNF-α: tumour necrosis factor alpha.

**Table 1 cells-09-00915-t001:** Neutrophil extracellular traps (NETs)-associated molecules that are known autoantigens in various autoimmune diseases.

Which Autoantigens Are Found on Neutrophil Extracellular Traps (NETs)?		To Which Autoimmune Diseases Are These Autoantigens Associated?	
α-enolase	[10,15]	SLE	[20]
Annexin A1	[10,15]	SLE	[21,22,23]
		RA	[22]
Apolipoprotein A1	[10]	SLE	[24]
Bb	[25]	AAV	[25]
C1q	[7,26]	SLE	[27]
Catalase	[10]	SLE	[28]
		RA	[28]
Cathelicidin	[10]	SLE	[21]
Citrullinated histones	[7,18,29,30,31,32]	RA	[18],
		SLE	[33]
dsDNA		SLE	[33,34]
Histones	[10]	SLE	[35]
HMGB1	[36,37]	SLE	[33,36,37]
LAMP-2	[38]	AAV	[39]
LL37	[5,7,10,40,41,42]	SLE	[41]
		Psoriasis	[43]
MMP8	[10]	RA	[44]
MMP9	[10,45]	SLE	[45]
MPO	[17,19,46,47,48,49]	AAV	[50,51]
PR3	[17,19,48]	AAV	[38]
Properdin	[25]	AAV	[25]
TF	[52,53]	SLE	[53]

Abbreviations: AAV: anti-neutrophil cytoplasmic antibodies (ANCA) vasculitis, Bb: complement factor b, C1q: complement component 1q, HMGB1: high mobility group protein B, LAMP-2: Lysosomal membrane 2 protein, MMP8: matrix metalloproteinases, MMP9: matrix metalloproteinase 9, MPO: myeloperoxidase, PR3: proteinase 3, RA: rheumatoid arthritis, SLE: systemic lupus erythematosus; TF: tissue factor.

**Table 2 cells-09-00915-t002:** Potential drug interventions targeting neutrophil extracellular traps (NETs) in autoimmune diseases (in vitro and in vivo evidences).

Potential Drug Interventions	Mechanism	Study Design	Effect on Neutrophil Extracellular Traps (NET)	Autoimmune Disease	Effect on Disease Clinical Endpoint	References
PAD4						
CI-amidine	PAD4 enzymes inhibitor	In vitro PMA-induced NETs	Blocked NET formation	AAV	Not tested	[90]
		Mouse model with MPO–ANCA production	Reduced citrullination and reduced serum MPO–ANCA level	AAV	Not tested	[90]
		SLE mouse model	Decreased histone citrullination and reduced release of NETs	SLE	Not tested	[91,92]
		pGIA mouse model	Reduced citrullinated proteins	RA	Reduced arthritis severity, but not significantly	[93]
GSK199	Low-calcium PAD4 enzymes inhibitor	In vitro ionomycin-induced mouse neutrophils	Inhibition mouse NETs	-	-	[94]
		In vitro *S. aureus*-induced human neutrophils	Partial NET formation remained	-	-	[94]
		Collagen-induced arthritis mouse model	Unknown	RA	Prevented clinical and histological disease severity	[95,96]
DNase						
DNase	DNA degradation	SLE mouse model	Unknown	SLE	Prolongation of survival	[97]
		Phase Ib study	Unknown	SLE	Unaffected disease activity	[14,98]
		APS IgG treated mice	Decreased NET formation	APS	Decrease in thrombus formation	[99]
		Anti-β2-GP1/β2-GP1 treated rats	Decreased NET formation	APS	Decrease in thrombus formation	[100]
ROS						
PRAK inhibitor	Inhibition of ROS-regulating PRAK	In vitro PMA-induced NETs	Increased neutrophil apoptosis over NET formation	-	-	[101]
Trolox	Antioxidant	In vitro PMA-induced NETs	Inhibited ROS-dependent NET formation	-	-	[102]
Tiron	Antioxidant	In vitro PMA-induced NETs	Remained NET formation	-	-	[102]
Tempol	Antioxidant	In vitro PMA-induced NETs	Inhibited ROS-dependent NET formation	-	-	[102]
Vitamin C	Unknown	In vitro PMA-induced NETs	Decreased NET formation	-	-	[103]
IFN-alpha						
Sifalimumab	Blocks IFN-α NETs stimulation	Phase I study	Unknown	SLE	Inhibited type I IFN signature and trend in improved SLEDAI	[104,105]
		Phase IIb study	Unknown	SLE	Improved SLE responder rate	[106]
Rontalizumab	Blocks IFN-α NETs stimulation	Phase I study	Unknown	SLE	No effect on IFN and anti-dsDNA levels	[107]
Anifrolumab	Blocks IFN-α NETs stimulation	Phase IIb study	Reduced plasma NETs complexes	SLE	Improved cholesterol efflux capacity	White et al., 2018 (unpublished, conference abstract)
		Phase IIb study	Unknown	SLE	Reduced SLE disease activity	[108]
Complement						
PA-dPEG24	C1 inhibitor	In vitro PMA, MPO or immune complex activated human sera	Inhibited complement activation and inhibited NET formation	-	-	[109]
Eculizumab	Antibody against c5a	PNH patients	Decreased neutrophil activation	-	Unknown	[110]
Proteases						
IcatC	CatC inhibition blocks PR3 activity and NET formation	Neutrophil differentiated CD34+ HSC	IcatC leads to absence of PR3 and suppression of PR3-ANCA antigen	-	-	[111,112]
Chloroquine	Inhibits autophagy	AP patients NETs	Decreased NET formation	-	-	[113]
		AP mouse model	Decreased NET formation and improved survival	-	-	[113]
Vitamin D						
Vitamin D	Unknown	In vitro PMA-induced NETs and endothelial cells	Reduced NET formation	SLE	Reduced endothelial apoptosis	[114]
Vitamin D	Unknown	SLE patients with low vitamin D	Unknown	SLE	Improved endothelial function	[115]
Nanoparticles						
A2,8-sialylated nanoparticles	Reduce PMA-initiated ROS production	In vitro PMA-induced NETs	Inhibited NET release	-	-	[116]
Polysialylated vesicles	Counteract cytotoxic characteristic of extracellular histones, possibly through lactoferrin	5B8 cells in presence of histones	Reduced cytotoxicity reduced in 5B8 cells. Vesicles bind to PMA–NETs	-	-	[117,118]
Existing autoimmune disease therapies and their effects on NETs						
-Tocilizumab	Antibody against IL-6 receptor	In vitro IL-6 and PMA-induced NETs	Blocked NET formation	RA	Unknown	[119]
Rituximab and belimumab	Blocking IC formation	Phase IIa study	Reduced NET formation	RA	Decreased lupus disease activity	[120]
IVIG-S	Mechanisms unknown	MPO-AAV rat model	Reduced NET formation and ANCA titers	AAV	Unknown	[121]

Abbreviations: AAV: Anti-neutrophil cytoplasmic antibodies associated vasculitis, ANCA: Anti-neutrophil cytoplasmic antibodies, AP: acute pancreatitis, APS: antiphospholipid syndrome, CatC: cathepsin C, CI-amidine: chloramidine, dsDNA: double strand DNA, EA: elastase-alpha1-antitrypsin, GSK199: hydrochloride, HSC: hematopoietic stem cells, IC: immune complex, IcatC: inhibitor cathepsin C, IFN: interferon, IVIG-S: sulfo-immunoglobulins, MPO: myeloperoxidase, NAC: n-acetylcysteine, NETs: neutrophil extracellular traps, PAD4: peptidylarginine deiminase 4, pGIA: glucose 6-phosphate isomerase induced arthritis, PMA: phorbol myristate acetate, PNH: paroxysmal nocturnal haemoglobinuria, PR3: proteinase 3, PRAK: p38-regulated/activated protein kinase, RA: rheumatoid arthritis, SLE: systemic lupus erythematosus, SLEDAI: systemic lupus erythematosus disease activity index, β2-GP1: β2-glycoprotein.

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
