# Peer review of "Neutrophil Extracellular Traps (NETs) Take the Central Stage in Driving Autoimmune Responses"

_cells, 2020, doi:10.3390/cells9040915_

Round 1

Reviewer 1 Report

  It is an interesting review article describing the central role of neutrophils extracellular traps in driving autoimmune responses. Accumulated evidence suggests that NETs may participate in the pathogenic mechanisms involving in autoimmunity, further contributing to the self-tolerance break in autoimmune diseases. The authors discussed the current findings on how NETs drive the autoimmune responses through being a source of autoantigens for autoantibodies and activating other immune cells like B cells, APCs and T cells. They concluded that NETs may be the central regulator of autoimmunity and serve as biomarkers as well as promising therapeutic targets for autoimmune disorders. 

  The manuscript is well written in English and the contents are relevant to the clinical application. There is no further comments on the manuscript. Accepted for publication is recommended by the reviewer.

Author Response

We are thankful to the reviewer for their positive feedback.

Reviewer 2 Report

The review article by Desai et al is an interesting discussion of the pathophysiology of NETosis (formation of neutrophil extracellular traps) and the role NETs play in autoimmune disease. This is a thoughtful, well-organized and relatively comprehensive review. However, I have a few concerns:

  1. My major concern is the lack of any discussion of the role NETosis plays in the pathogenesis of antiphospholipid syndrome (APS), particularly thrombosis in these patients. There has been quite a lot of work done on this subject in the last 5 years and the close association between SLE and APS indicates that a discussion of this topic would be extremely relevant to this review.
  2. The authors use several abbreviations throughout the manuscript without first giving their meaning
  3. There are only a few minor grammatical errors and misspellings throughout the manuscript but this should be reviewed.

Author Response

We are thankful to the reviewer for their positive evaluation. The queries raised by reviewer is addressed as below:

1. My major concern is the lack of any discussion of the role NETosis plays in the pathogenesis of antiphospholipid syndrome (APS), particularly thrombosis in these patients. There has been quite a lot of work done on this subject in the last 5 years and the close association between SLE and APS indicates that a discussion of this topic would be extremely relevant to this review.

Answer: Thank you very much for pointing out APS to be a very relavent autoimmune disease involving pathogy mediated by NETs. We agree and acknowledge that we did not include the relavent discussion in the previous version. We have now included the updated discussion in the manuscript (section 10.4).

2. The authors use several abbreviations throughout the manuscript without first giving their meaning

Answer: Abbreviation are defined now in the manuscript. The changes are marked as red in the revised manuscript

3. There are only a few minor grammatical errors and misspellings throughout the manuscript but this should be reviewed.

Answer: We have now corrected the minor spelling errors and typos throughout the manuscript.